# Sub-Wavelength Focusing in Inhomogeneous Media with a Metasurface Near Field Plate

**DOI:** 10.3390/s19204534

**Published:** 2019-10-18

**Authors:** Andrew C. Strikwerda, Timothy Sleasman, William Anderson, Ra’id Awadallah

**Affiliations:** 1Johns Hopkins Applied Physics Lab, 11100 Johns Hopkins Road, Laurel, MD 20723, USA; andrew.strikwerda@jhuapl.edu (A.C.S.); Raid.Awadallah@jhuapl.edu (R.A.); 2Johns Hopkins School of Medicine, 733 N Broadway, Baltimore, MD 21205, USA; wanders5@jhmi.edu

**Keywords:** sub-wavelength focusing, near field plates, diffraction limit, metasurface, evanescent spectrum, finite element method, corrugated surface, finite element method, sensing

## Abstract

Overcoming the diffraction limit, which enables focusing much less than the wavelength, requires tailoring the evanescent spectrum of an aperture’s field distribution. We model and simulate a corrugated near field plate, which can generate a sub-wavelength focus in inhomogeneous background media. All reactive coupling, between the metasurface near field plate and the focusing domain and among the corrugations in the metasurface, is taken into consideration with the finite element method, which we solve in combination with a constraint to generate a desired focus. Various geometries for the near field plate are considered and we demonstrate that the proposed method can effectively create a deeply sub-wavelength focus within a layered medium having properties resembling brain tissue. Such a device could find use as a detector of biological signals or for hyperthermic treatment near the skin surface.

## 1. Introduction

The ability to focus to a small spot size is invaluable when it comes to enhancing the resolution of imaging systems, increasing the sensitivity of detectors, and improving numerous other applications. Most systems encounter a hard barrier at the diffraction limit, which renders futile any attempt to recover spatial information significantly below the operational wavelength λ. However, there has been a longstanding notion that the diffraction limit can be circumvented at a stand-off distance using the near field of a system [1,2,3]. Interest in such possibilities has renewed in recent decades with the advent of the perfect lens, the superlens, and near field plates [4,5,6,7]. While achieved through different mechanisms, the common concept underlying these methods is tailoring of the evanescent spectrum of the imaging/sensing aperture so that scene components beyond the propagating wave spectrum can be probed. Resolution as small as λ/15 (or better) can be achieved by exploiting such techniques. Other approaches using nano-probes and apertures have achieved resolution of up to λ/1000 [2,3,8], but require placing a probe or aperture near the focal plane of interest and are currently unsuitable for real-time, non-invasive measurements in inhomogeneous environments.

It is often difficult to employ near field sensors because the probe couples with the entity of interest, altering its electromagnetic signature. Most systems concerned with near field focusing are therefore employed in free-space and have negligible coupling with the region of interest. Neglecting this coupling is detrimental to many applications where tight focusing is desirable in inhomogeneous settings, for example in wireless charging of biomedical implants [9] and local heating for ablation/hyperthermia treatments [10,11]. In such cases, it is necessary to account fully for the inhomogeneous environment when designing the focusing device.

Based on the notion of fields interacting with a region of interest, one can draw comparisons with the inverse scattering literature. In inverse scattering problems, the probing wavefields are often iteratively modeled by modifying not only the calculated image (be it a permittivity distribution or generic contrast), but also the Green’s function. The Green’s function captures how the fields propagate from one point to another, taking into consideration how the fields interact with the imaging domain [12]. This inverse scattering imaging problem is known to be ill-posed and is still an active area of research [13]. There has also been research into other iterative approaches to solving inverse problems, such as machine learning and the adjoint method that can result in appropriate designs with sufficient computational training [14,15]. Rather than considering the active imaging case (with both transmitters and receivers), here we concern ourselves only with transmission (for focusing/ablation) or reception (as in passive imaging). Additionally, we assume a priori information about the background media including its geometry and properties. While such a case seems restrictive, this situation can be expected in biomedical applications where magnetic resonance imaging (MRI) is used as a precursor to find the relative permittivity and conductivity throughout a voxelated volume. For example, in the context of neural ablation, the MRI data serves to inform the design of a focusing device. For a persistent sensor, the MRI data can give information on the background media in a large region, and then, the signals of interest (e.g., blood-oxygen level dependent (BOLD) signatures [16,17]) can be monitored in a subset of the volume.

A sub-wavelength focusing system for biomedical applications can take many form factors. A particularly promising architecture has been outlined by Merlin [6] and studied by Grbic et al. [7,18,19,20,21] in the form of near field plates (NFPs). Such devices are typically defined as tailored surfaces containing inductive and capacitive elements, which, through mutual interactions, sustain a field distribution that generates a focal spot [18]. Importantly, NFPs can be designed to operate at microwave frequencies, where a fair trade-off of penetration depth and resolution can be found for applications taking place in biological samples. Such NFP devices have been demonstrated for the generation of Bessel beams [22] and focal spots [18,20,21] in the evanescent near field. To date, NFPs have only been designed and demonstrated in homogeneous (free-space) settings based on semi-analytic propagation techniques in the region of interest. Further, in the past, the mutual interactions among the grooved elements in the surface have been computed analytically, which can lead to sensitivities and inaccuracies. For the mutual interactions and impedances based on lumped elements, more sophisticated models have improved the accuracy of these methods [21].

In this paper, we consider near field plates in inhomogeneous environments, a necessary extension for applications in biological sensing, imaging, and ablation. We begin with a corrugated metasurface NFP, as seen in Figure 1, that generates a focal spot with a full-width at half-maximum (FWHM) of λ/15. The structure is modeled via the finite element method (FEM) using a constraint imposed on the field in the focal plane. Whereas earlier methods made assumptions about interelement coupling and operated solely in free-space settings, our method takes full account of interactions among elements in the NFP surface and interactions between the NFP and the domain of interest. We begin with a planar two-dimensional (2D) simulation and then investigate axisymmetric geometries and also show an NFP realized with lumped circuit elements. The sensitivity of analytic assumptions and errors induced by non-homogeneous backgrounds are explored in detail. Finally, we implement a multiphysics model to study thermal effects and the possibility of utilizing NFPs in ablation application.

## 2. Design of Near Field Plates

Here, we will give a broad overview of some near field plate concepts and outline the traditional approach used to design them. We discuss the conventional analytic treatment and mention some potential pitfalls. The proposed model is then discussed, including a description of how the NFP’s geometry or surface impedance can be optimized using a constraint equation and Lagrange multipliers. This allows for sub-wavelength focusing in inhomogeneous settings, which is demonstrated in subsequent sections.

### 2.1. Underlying Concept and Analytic Treatment

We begin with a 2D geometry that is invariant in *z*, as depicted in Figure 1. It is assumed that the fields exhibit TM polarization, with nonzero magnetic field $H_{z}$ and electric field components in the xy plane. A focal plane is defined at a distance x=L away from an NFP’s surface. The NFP aperture plane at x=0 can contain reactive transmission line elements [7], lumped elements [21], or corrugations [20]. We restrict our discussion to the 2D geometry with grooves while explaining the concepts, but then delve into designs with lumped elements and axial symmetry in Section 4.

All grooves are shorted except for the central groove, which is used to excite the NFP with a transverse electromagnetic (TEM) plane wave. The groove width *w* and separation *a* are deeply sub-wavelength so that homogenized surface parameters can be used to model the NFP. Specifically, the surface impedance:
(1)$$Z\left(y\right)=\frac{M_{z}}{H_{z,NFP}}{|}_{x=0}$$
can be used to design the surface, where $H_{z,NFP}=H_{z}{|}_{x=0}$ is the total magnetic field in the plane of the aperture and $M_{z}$ is the equivalent magnetic surface current density. Note that the components considered here are based on our specific geometry and must be generalized for more complex systems.

The surface impedance gives us a means to design the magnetic field in the focal plane $H_{z,foc}\left(y\right)=H_{z}{|}_{x=L}$. Considering the fields in the spatial frequency domain, $H_{z,foc}\left(k_{y}\right)$ where $k_{y}=\sqrt{k_{0}^{2}-k_{x}^{2}}$, often provides insight for focusing applications. As discussed in [6], the choice of the pattern in the focal plane is essential to the design procedure. To overcome the diffraction limit, evanescent waves must be included in the support of $H_{z,foc}\left(k_{y}\right)$, and the domain of the transverse wavenumbers must therefore exceed the free-space wavenumber $k_{0}$.

For the field/source distribution along the aperture plane, it is ideal to have negligible contributions in the propagating portion of the spectrum $\left|k_{y}\right|<k_{0}$. As seen in Figure 2, confining the plane waves in the source plane to the evanescent regions $k_{0}<k_{1}<\left|k_{y}\right|<k_{2}$ is desirable to mitigate radiation. If the spatial frequency spectrum in the focal plane is given as a rectangular function, e.g., $H_{z,foc}\left(k_{y}\right)\propto \mathrm{rect}\left(k_{y}/\left(2k_{2}\right)\right)$, then the focal pattern will take the form $H_{z,foc}\left(y\right)\propto \mathrm{sinc}\left(k_{2}y\right)$. Using these functions in a specific example, the support of $H_{z,foc}\left(k_{y}\right)$ should extend to $k_{2}\approx 6k_{0}$ to achieve a λ/10 FWHM in the spatial domain. Examination of the necessary source distribution to achieve such a pattern, as was done in [18], reveals that there are negligible fields in the propagating band at the NFP surface. To remain consistent with comparable works in the literature, here we will define a desired magnetic field $H_{theory}=\mathrm{sinc}\left(qy\right)$ where $q=10k_{0}$ in the focal plane and use this as the basis for designing the NFP.

Creating the desired pattern in the focal plane can be achieved by tailoring the effective magnetic current density in the NFP plane. The focal plane field is related to these effective sources by:
(2)$$H_{z,foc}\left(y\right)=\int_{{\bar{r}}^{\prime }\in \mathrm{NFP}}M_{z}\left({\vec{r}}^{\prime }\right)G\left({\vec{r}}^{\prime },\vec{r}\right)d{\vec{r}}^{\prime },\;\mathrm{with}\;\left\{\vec{r}\in \left(x,y\right):x=L\right\},$$
where $G({\bar{r}}^{\prime },\bar{r})$ is the Green’s function from a point on the aperture ${\bar{r}}^{\prime }$ to a point in the focal plane $\bar{r}$. In practice, the grooves in the NFP can be discretized and are summed over a finite extent containing *N*-many grooves. Assuming a time-harmonic variation of the form exp(jωt), the Green’s function in a homogeneous background media is analytically given by the zeroth order Hankel function of the second kind, $H_{0}^{\left(2\right)}$. In this case, Equation (Equation 2) turns into:
(3)$$H_{z,foc}\left(y_{m}\right)=\frac{-\omega \epsilon_{0}}{2}\sum_{n\in N}M_{z}\left(y_{n}^{\prime }\right)H_{0}^{\left(2\right)}(k_{0}\sqrt{{\left(y_{m}-y_{n}^{\prime }\right)}^{2}+L^{2}})$$
where ω is the angular frequency and $\epsilon_{0}$ is the vacuum permittivity. Inserting $H_{theory}$ for $H_{z,foc}$, this equation can be inverted to find the necessary effective sources. Certain geometries admit exact solutions for $M_{z}$ [6], but most scenarios require numerical techniques such as the method of moments [20].

With a target $M_{z}$ known, we can now find the total field along the NFP aperture $H_{z,NFP}$. This total field will be the summation of the incident field with the fields from the desired $M_{z}$ sources. Extra attention must be given to the center slot (at y=0) to normalize the fields/currents correctly and achieve a good impedance match; this is discussed in detail in [20], but is not critical to the conceptual understanding, and we omit these details for brevity. Here, we simply normalize the incident excitation field at the center slot to be a constant magnitude. Knowing the desired set of sources from the inversion of Equation (Equation 2), we put these sources back into the same equation with the modified condition $\left\{\bar{r}:x=0\right\}$. Similarly, Equation (Equation 3) is updated by setting L=0. Once the total field $H_{z,NFP}$ is found, it can be substituted into Equation (Equation 1) with $M_{z}$ to find *Z*. The design procedure then becomes finding the geometry that achieves the necessary surface impedance.

The approach to determine the groove depths is akin to other metasurface design procedures, such as modeling the surface as an effective susceptibility [23,24] or as a collection of discrete dipoles [25,26,27]. There is also a large body of literature on the impedance of corrugated surfaces [28,29]. Previous approaches employed to design corrugated NFPs have modeled the grooves as shorted transmission lines. The input impedance of the transmission line, which is equivalent to the surface impedance in our case, is given by:
(4)$$Z_{in}\left(d\right)=jZ_{0}\tan\left(k_{0}d\right)$$
for a groove with depth *d*. While this model, shown as the black, dotted line in Figure 3, is reasonable, it is based on some approximations. For instance, it does not take into consideration the mutual interactions between the slots. Additionally, when the aperture is finite and not perfectly periodic, the impedance model will start to have greater deviations from any analytic estimation. While these discrepancies seem to have small effects, it turns out that the desired groove depths are very close to the resonant length of the transmission line (λ/4) and the asymptotic behavior of the tangent function in Equation (Equation 4), which dramatically increases the sensitivity of the design process.

Even in the purely periodic case with deeply sub-wavelength spacing, if the structure is simulated, there is a discrepancy between the model in Equation (Equation 4) and the extracted surface impedance (based on the method in [23,30]). This difference is plotted in Figure 3 for a variety of periods. For a deeply sub-wavelength spacing of λ/40, there will be a deviation of 20% between the analytic expression and the extracted impedance at d=0.243λ. In several earlier works [21,22], it was seen that the operating frequency of the designed NFP device had to be slightly detuned from the intended frequency, a behavior that may be caused by this type of discrepancy. Since many of the slot lengths fall into the 0.24λ–0.25λ range, the tangent curve can be slightly shifted (equivalent to frequency detuning) to create a better match. We will explore this possibility in Section 3.

Finally, it is worth noting that the impedance in Equation (Equation 4) is purely imaginary, while the desired impedance from Equation(Equation 1) will have some real components. This will lead to further discrepancies.

In addition to these approximations, it is worth mentioning that all of this analysis is contingent on free-space operation, whereas our desired application requires the consideration of inhomogeneous media in the focusing domain. Within the above analysis, the Green’s function in Equations (Equation 2) and (Equation 3) was based on the analytic solution in free-space, and the groove lengths were designed based on Equation (Equation 4). Both of these restrictions represent drawbacks for our application. To address these shortcomings, we modify the design process, which is outlined in the next subsection.

### 2.2. Rigorous FEM and Lagrange Multiplier Model

We implemented a numerical model to address the the shortcomings outlined in Section 2.1. Starting from a known geometry and associated material properties, this model solves for the electromagnetic fields throughout the domain and accounts for the physical implementation of the desired impedance profile of the NFP. This design methodology directly accounts for the inhomogeneous geometry and material properties in the focusing domain (e.g., skull and brain tissue) and any modifications to the NFP implementation due to mutual coupling between elements or coupling between elements and the dielectric domain. We implemented this model using the commercial numerical package COMSOL Multiphysics, but emphasize that this is not an endorsement, but rather an explanation of the method implemented in this work. We expect that other commercial solvers or home-built code could accomplish a similar implementation.

In this model implementation, we directly solve for the physical parameters of interest by adding a constraint equation and a number of Lagrange multipliers. If the design of interest is a corrugated sheet, the Lagrange multipliers correspond to the depths of individual grooves. Later, we use lumped circuit elements to implement the needed surface impedance profile, and in that case, the Lagrange multipliers will instead represent the impedance of the lumped elements. Regardless of the physical implementation, we need one multiplier per unique element. Using the nomenclature of the calculus of variations, the constraint equation is represented as:
(5)$$g(\vec{r},u,u^{\prime })=0$$
where *u* is the function to be optimized and $u^{\prime }$ is its spatial derivative. Since FEM itself can be thought of as a variational problem, adding the Lagrange multiplier method is a natural extension [31]. Here, we focus solely on the constraint handling and how to incorporate it with the commercial solver. Upon taking the variation of the constraint over the simulation domain, we have:
(6)$$0=\delta \int_{\Omega }\Lambda_{n}g\left(\vec{r},u,u^{\prime }\right)dV=\delta \Lambda_{n}\int_{\Omega }g\left(\vec{r},u,u^{\prime }\right)dV+\Lambda_{n}\int_{\Omega }\left(\frac{\partial g}{\partial u}\delta u+\frac{\partial g}{\partial u^{\prime }}\delta u^{\prime }\right)dV$$
where $\Lambda_{n}$ is a Lagrange multiplier (note that we use λ for the wavelength and Λ for the Lagrange multiplier to avoid ambiguity). It is then left to incorporate this equation into the simulation. In our case, we are interested in specifying a desired field, so that the constraint equation of interest is $g\left(\vec{r},F\right)=F_{Theory}-F=0$, where $F_{Theory}$ and *F* are the desired theoretical and simulated fields, respectively, and could represent either an electric or magnetic field. To enforce the constraint’s domain of applicability, we insert a Dirac delta function into the integrands so that the integration domain is reduced to the focal plane (FP). For the second term, we see that $\frac{\partial g}{\partial F}=1$, and the integral evaluates to a constant that can be neglected. This reduces Equation (Equation 6) to:
(7)$$0=\delta \Lambda_{n}\int_{FP}g\left(\vec{r},u,u^{\prime }\right)dS+\Lambda_{n}\delta F.$$


At this point, we have reduced the constraint to a form that can be readily incorporated into the software.

### 2.3. Model Implementation

We begin by discussing the specific model implementation shown in Figure 1b, which models a 2D corrugated NFP, and later extend our discussion to 2D axisymmetric geometries and lumped circuit elements. In any of the model implementations, we want to determine the physical parameters that achieve the desired field profile in the focal plane using the Lagrange multiplier method described in Section 2.2. In this particular implementation of a corrugated sheet, that means determining the groove depths and magnitude of the source excitation, and the focal plane is the vertical line in Figure 1b. Since this is a 2D model, we assumed translational invariance in the out-of-plane direction. To compare with previous work, we chose a desired field of:
(8)$$H_{theory}\left(y\right)=\mathrm{sinc}\left(qy\right)$$


For clarity in the following section, we will explicitly mention the software features used, using the COMSOL software terminology, and indicate these with italics. We assume that the corrugated sheet is a *perfect electric conductor* (PEC), and this boundary condition is applied to the surfaces bounding the central waveguide, the 38 grooves, and the remainder of the metallization at x=0. Note that the PEC could be replaced with a lossy metal, such as copper, to characterize more accurately a particular realization. The outer layer of the hemispherical domain is a *perfectly matched layer* (PML) that truncates the simulation by absorbing any outgoing waves without reflection. Because of symmetry and the polarization of the desired field, the long central groove is a 2D waveguide excited by a TEM mode. Since we do not know the magnitude of the incident wave a priori, we used *global equations* to add a Lagrange multiplier, $\Lambda_{0}$, to the simulation and used this variable as the magnitude in a *surface magnetic current density* boundary condition. This pre-existing boundary condition handles the second term on the right-hand side of Equation (Equation 7), and we incorporate the first term using a *weak contribution* on the focal plane. This weak contribution allows us to add equations directly to the variational problem the software is solving. In this case, we added $\left(H_{theory}-H\right)*test\left(\Lambda_{0}\right)$, as we needed to replace variational operator δ with the *test()* function incorporated in the software.

The grooves shown in Figure 1b have a depth of λ/4, which is their initial value, but are deformed during computation to achieve the desired field in the focal plane. We implemented this by adding a *deformed geometry* interface. The open face of the groove at x=0 is fixed in place using a *prescribed mesh displacement* boundary condition. The remaining boundaries are subject to the same boundary condition, but are allowed to move in *x*, but not *y*, so that they can stretch and compress while the width of the groove remains constant. Lastly, the displacement of the groove bottom is specified by $\Lambda_{n}$, where *n* corresponds to the groove number. As before, the pre-defined boundary condition satisfies the second term of Equation (Equation 7), and we incorporated the first term through the addition of another *weak contribution*.

Through the incorporation of the constraint and Lagrange multipliers into the finite-element method, we reduced the optimization problem to a single nonlinear matrix equation Ax=b. In solving the design problem in this way, there are two points that are worth mentioning explicitly. The first point is that care must be taken when setting up the nonlinear problem, as the solution convergence can depend significantly on initial conditions and solver settings. As a specific example, the simulation in Section 4 is solved readily when focusing on a homogeneous environment, but not so in the inhomogeneous environment that we demonstrate. As a result, we multiplied the inhomogeneous material properties by a dummy variable that we scaled from 0 to 1 over the course of the simulation. In this way, we used the homogeneous solution as an initial guess and slowly “walked” the solution to the desired inhomogeneous environment to achieve model convergence. The second point is that we expected this method to scale extremely well with regards to the number of unknown design parameters. This is because each additional Lagrange multiplier will only add one additional row to the matrix equation, whereas more traditional parameter optimization techniques generally scale poorly with the number of parameters. While we have not done a quantitative comparison against other optimization techniques, the 2D model in Section 3 was computed in 6 s, and the 2D axisymmetric models in Section 4.1 and Section 5 took 36 s and 7 s, respectively, and the 3D sector model in Section 4.2 taking 2 min 15 s. All of these were computed on a dual socket 2x8 core Intel Xeon E5-2630 @ 2.40 GHz.

## 3. Results for the 2D Planar Geometry

As a first study, we looked at a near field plate design and its resulting fields in free-space. We compared the proposed technique with the previously developed analytic framework. The planar and grooved NFP was employed at a frequency of 1 GHz. A total of 19 grooves appeared on each side of the central feed. The width of each groove was w=λ/80, and they were spaced with a periodicity of a=λ/40. A focal length of λ/15 was selected, and the target field was $H_{theory}=\mathrm{sinc}\left(10k_{0}y\right)$.

For the analytic case, we used two different models for the surface impedance based on Equation (Equation 4). We employed a strict application of this model and also considered a slightly tweaked version, which was similar to the frequency detuning mechanisms used in [20,21,22]. The frequency detuning is akin to shifting the input impedance defined in Equation (Equation 4). The amount of shift required is determined by matching the extracted surface impedance in Figure 3 with the analytic tangent function. This is achieved by adding a constant offset, equal to 0.011, to the argument of the tangent function. Note that we shifted by a positive amount because most of the groove lengths fall between 0.24λ and 0.25λ. The argument of the tangent function can be refactored as:
(9)$$k_{0}d+0.011=d\left(k_{0}+\frac{0.011}{d}\right)=\frac{2\pi d}{c}\underset{f^{\prime }}{\underset{︸}{\left(f_{0}+\frac{0.011c}{2\pi d}\right)}},$$
where $f^{\prime }$ is the equivalent detuned frequency. For a groove of length 0.242λ, this amounts to an approximate 0.7% shift in the frequency, a value commensurate with the result reported in [20]. While this phenomenon was not discussed in detail in [20], the explanation and result reported above seem to explain the behavior as seen in previous works. The shift of 0.7% may seem minor, but, as seen in Figure 4a, it can have a dramatic effect. Note that when following this process, the focus occurs at the true design frequency, 1 GHz, rather than having to tweak by an arbitrary offset, as was done in [20]. Operating at the true design frequency allowed us to make a more fair comparison with the proposed method that we have introduced in Section 2.2.

By employing the FEM with Lagrange multipliers, we arrived at the desired focus without any of the nuances or tweaks required for the analytic method. The result is also shown in Figure 4a and is seen to have better agreement with the theory than either of the other results. For all three cases, we also plotted the lengths of the grooves in Figure 4b. In this plot, it can be seen that the differences between the three methods are minor, even though the focusing result is starkly improved. It is also interesting to note that the proposed method has groove lengths that lie in between the two other methods, but the grooved pattern cannot be explained as a constant offset from the others. In summary, the presented approach improved on previous methods and also allowed for the extension to inhomogeneous cases, as we show next.

To demonstrate focusing in an inhomogeneous setting, we place a pair of dielectric slabs with refractive index n=2 in the domain, as shown in Figure 5a. The slabs had a length of λ/30 in *x* and λ/4 in *y*. They were offset by a variable distance ∆y, shown in Figure 5a, which is swept from a value of λ/2 to a value of 0. For some of these separation distances, the blocks were overlapping, and the intersecting area maintained the same refractive index. Figure 5b,c shows two examples for the block placements and the fields throughout the domain. Note that the groove depths changed for the different block locations to maintain the desired field in the focal plane.

Updating the groove design for the different block locations allowed us to maintain the focus for a variety of different inhomogeneous settings. In contrast, in the analytic approach the grooves did not depend on the environment. In this case, the focus deviated from the desired pattern, creating different field magnitudes and altering the side lobe levels. These trends can be seen in Figure 6: in Panel (a), the grooves are left as the free-space design would dictate, and in (b), the grooves are updated according to the proposed method. As can be seen, the Lagrange multiplier method allowed us to maintain the desired field profile regardless of the dielectric objects in the focusing domain. The results in Figure 5 and Figure 6 represent snapshots in a simulation, which can be found as a Appendix A in the online version of this article.

## 4. Flat, Axisymmetric Near Field Plate

In the previous section, we assumed a geometry that extends infinitely in the *z* direction. To create a physically-realizable version, such a device would need to be truncated to create a finite aperture. An alternative geometry that is worthy of consideration is an axisymmetric structure, which was pursued in [21,22]. Here, we will look at a groove-based geometry (similar to Section 3) and also a geometry based on a printed circuit board (PCB). Minor differences exist between the two, but it should be kept in mind that both grooves and PCB lumped elements can be utilized to create a desired impedance profile, which is the crux of the method outlined in Section 2. For reference, we will use the cylindrical coordinate system (ρ,ϕ,z).

### 4.1. NFP with Grooves

The first axisymmetric geometry closely resembles the corrugated infinite structure (see Figure 7), but the flat grooves instead take the form of a concentric circle around a central feed. A two-dimensional simulation can capture the field profile in this setting with the condition that the fields have no variation in the ϕ dimension. More sophisticated models may allow for variation of E(ϕ), H(ϕ), and Z(ϕ), but since we will be considering an axisymmetric focus, it is not necessary to include the higher order modes needed to capture this behavior. As with the infinite geometry, the groove depths define the impedance. Further, the groove depths are optimized with the constraint as outline in Section 2 rather than with the analytic model in Equation (Equation 4).

Just as a desired field pattern $H_{theory}=\mathrm{sinc}\left(qy\right)$ was prescribed for the previous geometry, here we identify a focal plane at a given *z* and define a target pattern. In this case, we look at the electric field oriented along the *z* axis and target the pattern $E_{z,theory}\left(\rho \right)=J_{1}\left(8\pi k_{0}\rho \right)/\left(8\pi k_{0}\rho \right)$, where $J_{1}$ is the Bessel function of the first kind, in the plane z=λ/15. This field only depends on the radial location because it is defined in a single *z* plane and has no variation in the angular dimension. Note that the field is directed along the axis of symmetry. Alternative designs for near field focusing have considered the creation of transversely-polarized fields at the focal spot (for example, for λ/2 focusing the in the radiative near field in [32]); addressing this possibility remains an interesting path of future study for our method.

For this geometry, we considered three different scenarios. In the first case, we designed an NFP to focus in free-space. In the second case, we added a representative biological body centered on the focal plane and then designed the NFP to focus at the center of this inhomogeneous environment. In the third case, we used the NFP designed in Case 1, except that we applied it to the inhomogeneous environment used in Case 2. This was done to determine whether the proposed design method that accounts for inhomogeneous environment (Case 2) outperformed the naive design approach (Case 1), which did not consider an inhomogeneous setting. The biological entity was composed of tissue that would be found in the human head. Specifically, there were two layers with the inner radius being 2 cm and a thickness of 1 mm for the outer layer. The outer “skull” layer had $\epsilon_{skull}=20.6$ and $\sigma_{skull}=0364$ S/m. The inner “brain” layer had $\epsilon_{brain}=48.9$ and $\sigma_{brain}=1.31$ S/m. These values, which were taken from the IT’IS database [33], were used as a proof of concept demonstration in this paper, but more appropriate sizes and properties could be incorporated based on the application at hand.

The results for the three different cases are compared with the targeted focal field in Figure 7b. It is seen that the NFP that is optimized for a free-space setting (Case 1) operates well when utilized in free-space, but once the biological tissue is included (Case 3), the focus is completely annihilated. For Case 2, where we optimized the NFP with the tissue taken into consideration, the result matched the desired field extremely well. Some deviations can be seen at ρ/λ=0.04, which was where the interfaces/discontinuities occurred in the material. The complete breakdown of the focus when the inhomogeneous setting was not considered constituted a clear demonstration that our method was successful and necessary for such environments. A final isometric plot in Figure 7c shows the fields in the grooves and in the volume, as well as the Ohmic heating ($\vec{J}\cdot \vec{E}$) that would occur within the targeted volume.

### 4.2. NFP with Printed Circuit Board

The grooved structure explored in the previous structure was cumbersome, heavy, and would not be cost-effective for bulk fabrication. Instead, a favorable architecture can be found in PCB fabrication processes [21,34]. A planar PCB with copper cladding on both sides can be fed at the center to create a cylindrically-symmetric parallel plate waveguide mode. Cutting concentric annular slots into the top copper layer of this PCB results in leaked fields that can be focused. The concentric annular slots represent capacitive slots in the transmission line, but can generally be viewed as impedances if lumped components (such as capacitors and inductors) are used to load the device. Designing the impedance profile through this mechanism gives a route to tailoring the effective sources across the aperture, which enables the desired sub-wavelength focusing.

A semi-analytic method to design such PCB-based NFPs was developed in [21], but suffered from similar drawbacks to those outlined in Section 2. Specifically, the model was incapable of handling inhomogeneous background environments. Further, the model can suffer from sensitivities and inaccuracies when operating near the elements’ resonance frequencies. In our treatment, the constraint and Lagrange multipliers that were previously applied to the depth of the grooves were now associated with the impedance of the loading lumped components. A reactive impedance model was used, which can be realized by way of a series of parallel LC circuit attached across each annular slot.

In contrast to the groove-based geometry, the PCB architecture was simulated as an angular section covering a range of 0<ϕ<π/8 (see Figure 8). This had the effect of creating 16 lumped components per annular ring that were uniformly placed along the ϕ dimension. The slots were separated by a distance a=6 mm and had a width w=0.4 mm, and the overall radius of the device was R=42 mm. The same focal distance was used, and the desired field was a truncated Bessel function, $E_{z,theory}=\exp\left(-\rho^{2}/2\sigma^{2}\right)J_{0}\left(q\rho \right)$, where σ=23 mm and $q=7.6k_{0}$. The match between theory and simulation is shown in Figure 8b. It can be seen that there were minor deviations beyond a distance of ρ/λ=0.14. This deviation was attributed to the fact that this radial distance fell outside of the circumference of the NFP. Finally, the fields in free space can be seen in Figure 8c. Such a device can also be fabricated for use in an inhomogeneous setting, but this result was omitted for brevity.

## 5. Curved, Axisymmetric Near Field Plate

The methods above can be further generalized to alternative geometries, including arbitrary shapes or structures that are conformal to the desired target volume’s surface. For example, one can imagine placing an NFP over a patient’s head to ablate a tumor within neurological tissue. In this setting, it may be advantageous to have a curved device that fits comfortably on the subject.

To show that the above techniques and concepts can be generalized to other geometries, we introduced the NFP in Figure 9, which is both axisymmetric and curved. Because neither the grooved geometry nor the PCB geometry in Section 4 are easily defined/fabricated in this geometry, we restricted ourselves to only defining the necessary phi-directed effective magnetic current density $M_{\phi }$. In this sense, we omitted the step of calculating the impedances in Equation (Equation 1). The radius of curvature of the conformal surface can be modified while maintaining the same focusing pattern within the region of interest. The results for a variety of radii are shown in Figure 9a–d. The fields in the focal plane are also shown in Figure 9e. For this study, we reused the same focal pattern as for the axisymmetric corrugated plate, $E_{z,theory}\left(\rho \right)=J_{1}\left(8\pi k_{0}\rho \right)/\left(8\pi k_{0}\rho \right)$.

In Figure 9e, it can be seen that the pattern for the 5-cm-radius case significantly deviates from the target function. As can be seen in Figure 9d, this is because the focal plane is located in close proximity to the curved surface. To circumvent this issue, one could imagine a focal curve rather than a focal plane. This curve might be defined to be concentric with the curved NFP geometry or the curve could follow an arbitrary path. It is also worth noting that there is no guarantee that the desired pattern will be achievable, as it could be possible that the target pattern has significantly higher spatial frequency components than are physically realizable. It may also be possible that the necessary mutual interactions among the tunable impedances may not support the necessary field that generates the desired effective sources $M_{\phi }$ (and therefore, the target field). Investigations of these limitations are left to future works.

## 6. Heating Effects and Ablation

When considering near field plates for an ablation application, an additional study of interest is to observe the heating profile of such devices. To do this, we employed a COMSOL Multiphysics model including both EM and heating effects. The heating distribution is taken into consideration by using the bio-heat equation, which models the specific heat, thermal conductivity, metabolic heat, and other effects [10]. For simplicity, we reverted to the 2D planar geometry and included a background media resembling biological tissue. A tumorous body (radius = λ/80) was also included with different material properties. All properties (EM, heat-related, and biological, as seen in Table 1) were taken from [10], which studied the ablation of breast tumors based on an ultra-wideband, time-reversed, far field array of antennas. The approach pursued by Converse et al. in [10] represents a promising alternative to our approach, but can be more costly due to the numerous antennas and the wide bandwidth required [35,36]. To simplify our study and obtain a focused electric field, we modeled the NFP surface as effective electric currents (rather than a magnetic current). We applied a temperature boundary condition along the NFP surface such that the neighboring region remains at a temperature achievable with cold water (3 °C). For the heating module, the *z* domain is truncated to model the volume as a 30 cm-thick slab rather than an infinite volume.

The result of this simulation is shown in Figure 10. The temperature in the focal plane was warmest near the tumor, as expected, as that is where the magnitude of the focused electric field was greatest. However, it is also seen that there was a larger temperature spike within the volume at a distance of 1 cm, whereas the tumor was located at 2 cm. This is because the NFP inherently created an exponentially-decaying field along the *x* direction. Such behavior is physically necessary for the generation of the sub-wavelength focused evanescent fields. Based on this heating behavior, it seems likely that such a technique would only be successful at ablating tissue located in close proximity to the surface. It can also be concluded that a sensing/detection or power transfer application may be a better fit for this technology [37,38]. A potential application could be monitoring a sub-wavelength region within a larger volume to detect changes in electromagnetic properties. This eliminates the concerns associated with heating non-targeted tissue since the fields are of a significantly lower intensity.

## 7. Conclusions

In this paper, we explored the use of microwave near field plates in the context of inhomogeneous settings. Deeply sub-wavelength focuses were demonstrated numerically in both free-space and inhomogeneous environments. Whereas previous methods only considered analytic or semi-numerical approaches, our method took the complete geometry of the system into account to design the NFP. We included the surface impedance of the NFP directly into a finite element method solver, which can compute the necessary geometry or lumped components to create the desired pattern. This method was shown to generate the desired focus in inhomogeneous environments, and for comparison, we showed that a design optimized for free-space operation, when applied to the inhomogeneous environment in question, resulted in a decimated focus.

The proposed method was applied to a variety of geometries, including 2D planar, flat axisymmetric, and curved axisymmetric. Biological bodies resembling neurological matter and breast tissue were investigated, and heating inside of a biological media was studied. Such a device could be used for sensing/detection or for ablation applications if the target is near the surface. Our method presents a viable option for the future development of near field plates, opening up potential routes to using such devices in practical inhomogeneous settings.

## Figures and Tables

**Figure 1 sensors-19-04534-f001:**
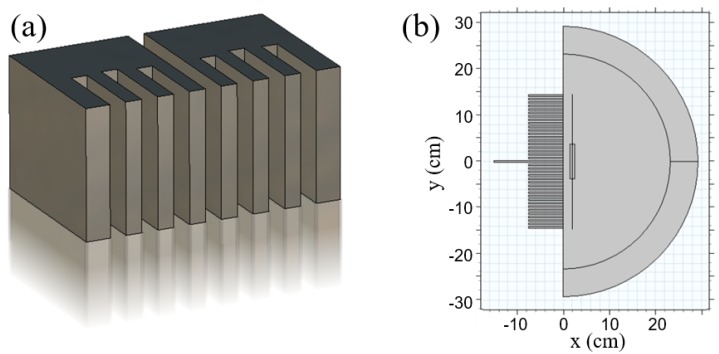
(**a**) A schematic depiction of a planar near field plate with a center feed and flanking grooves. (**b**) The geometry utilized in the finite element method simulations, with infinite extent in the *z* axis, a focal plane at an offset *x*, and inhomogeneous media in the region of interest. The groove width, spacing, and length are 3.7 mm (λ/80), 7.5 mm (λ/40), and 74.9 mm (λ/4), respectively, at a frequency of 1 GHz.

**Figure 2 sensors-19-04534-f002:**
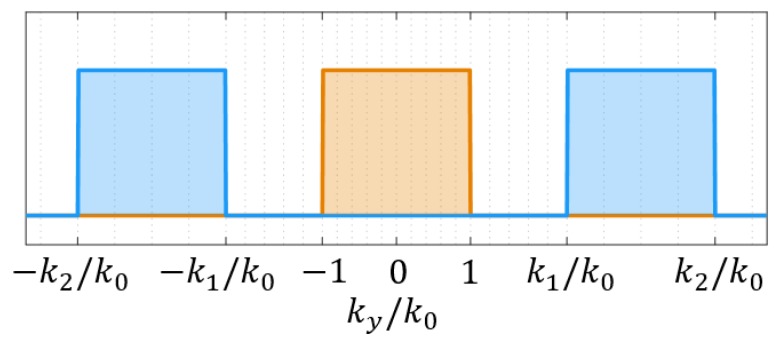
The desired portions of the normalized spatial frequency spectrum are highlighted as the blue regions. To mitigate radiation, there should be negligible contributions in the propagating wave region, highlighted in orange.

**Figure 3 sensors-19-04534-f003:**
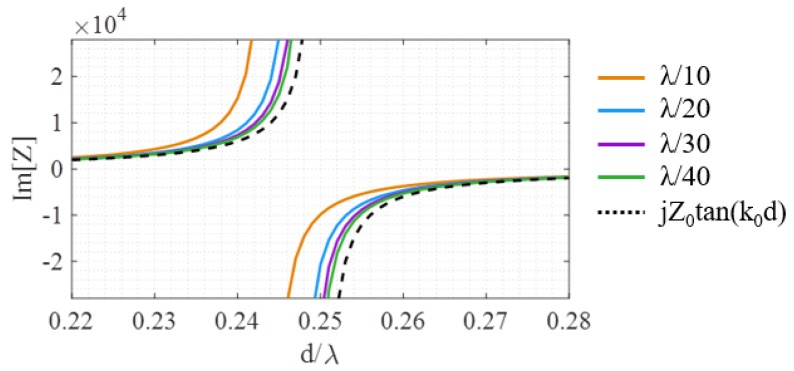
The imaginary part of the homogenized effective impedance of the surface when loaded with corrugations of given periodicities. As the period decreases from λ/10 to λ/40, the result approaches the analytic model, but the error remains non-negligible near resonance. The width of the groove is half of the periodicity.

**Figure 4 sensors-19-04534-f004:**
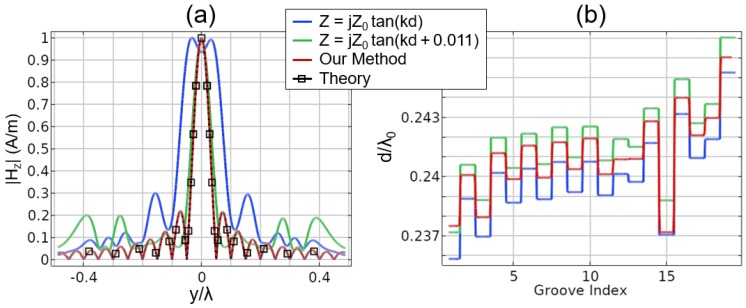
(**a**) Focusing results for three different methods and their comparison to the theory. (**b**) The associated groove depths for each model. Blue represents the pure analytic model; green is a minor tweak to better fit the extracted impedance result; and red is the proposed method that includes the groove depth as part of the FEM process. All results are in free-space.

**Figure 5 sensors-19-04534-f005:**
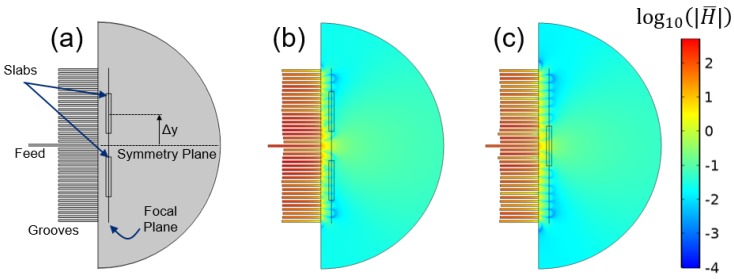
(**a**) Geometry for the inhomogeneous focusing study, with two blocks having n=2. (**b**,**c**) Spatial distribution of the magnetic fields, $\log_{10}\left(|H_{z}|\right)$, for ∆y=0.65 and ∆y=0, respectively. Note that the groove depths have changed, but the focused field remains the same.

**Figure 6 sensors-19-04534-f006:**
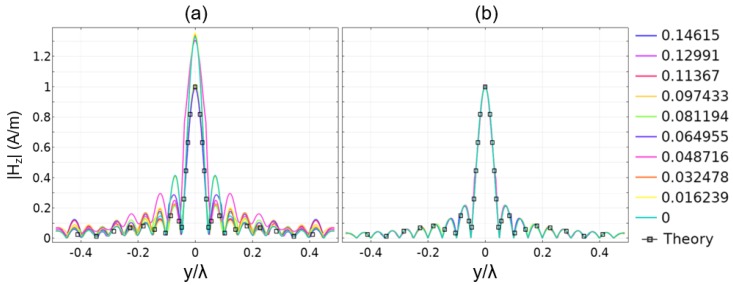
Profile of the magnetic field in the focal plane for a variety of separation distances for the geometry in Figure 5. (**a**) Without correcting for the contribution of the dielectrics and (**b**) correcting for the full domain with the constraint imposed in the FEM model.

**Figure 7 sensors-19-04534-f007:**
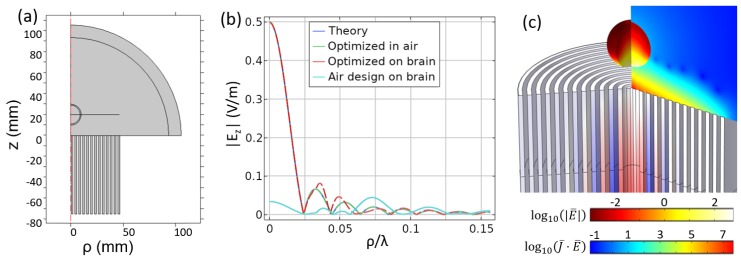
(**a**) A slice of the axisymmetric geometry, (**b**) the field in the focal plane, (**c**) and full field results. For the isometric view, we show Ohmic heating, $\log_{10}\left(\vec{J}\cdot \vec{E}\right)$, in the biological tissue; a slice of the electric field, $\log_{10}\left(|\vec{E}|\right)$, in the air between the grooves and target; and a measure of the H-field, $\rho \cdot H_{\phi }$ in the grooves. Note that the color bars are Ohmic heating (left) and E-field (right), but we do not show a color bar for the grooves so as not to clutter the figure. The relevant H-field information is that the phase oscillates from groove to groove and the amplitude vs. radius decays faster than linear.

**Figure 8 sensors-19-04534-f008:**
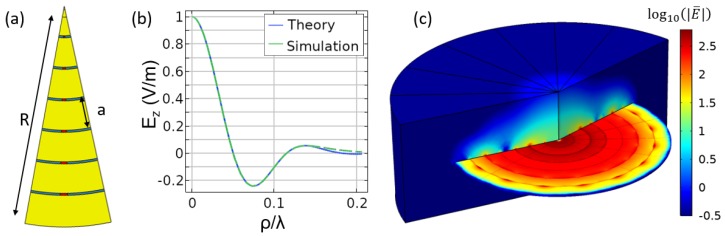
(**a**) The geometry for the PCB-based NFP, with only a single sector comprising 1/16 of the surface shown. (**b**) A free-space result comparing the desired and resulting field profile. (**c**) Logarithmic plot of the electric field, $\log_{10}\left(|\vec{E}|\right)$, in the NFP and focusing domain.

**Figure 9 sensors-19-04534-f009:**
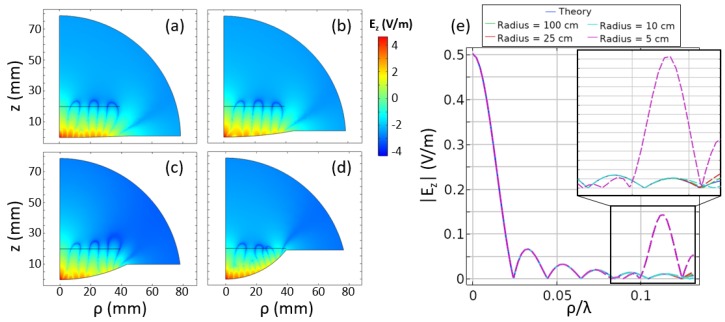
(**a**–**d**) The geometry and electric field for various radii, including 100 cm, 25 cm, 10 cm, and 5 cm, respectively. (**e**) The electric field in the focal plane, which generally exhibits a good match between the desired theory and the achieved pattern. Notable deviation is seen in the case of the 5-cm radius because of the close proximity between the focal plane and the designed surface.

**Figure 10 sensors-19-04534-f010:**
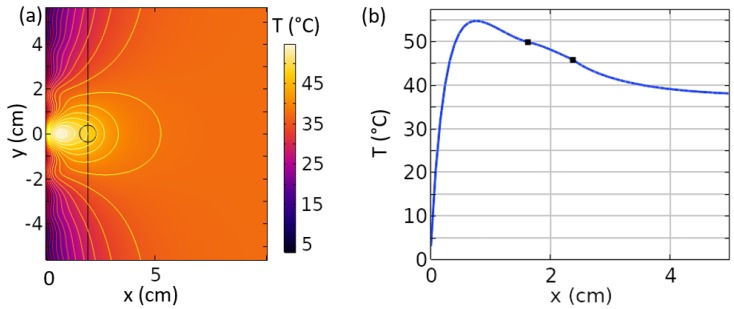
(**a**) Temperature distribution within a biological volume when attempting to ablate tumorous tissue with an NFP. The focal plane is represented by the black line at x = 2 cm, and the tumor outline is shown centered at x = 2 cm, y = 0. (**b**) A plot of the temperature along the line y=0 with the boundaries of the tumor marked by black dots. Maximal heating does not occur within the tumor, but is generated at a shallower depth due to the evanescent nature of the fields.

**Table 1 sensors-19-04534-t001:** Electromagnetic and heating properties used in the heating simulation. Both regions share the same properties for: arterial blood temp = body temperature, specific heat of blood = 1 J/(kg·K), blood perfusion rate = 1 Hz, and density of blood = 1000 kg/m^3^.

	Background	Tumor
Relative Permittivity	15.66	50.74
Electrical Conductivity, S/m	1.03	4.82
Density, kg/m^3^	1069	1182
Heat Capacity, J/(kg·K)	2279	3049
Thermal Conductivity, W/(m·K)	0.306	0.496
Metabolic Heat Source, W/m^3^	350	5500

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
