# Peer review of "Sub-Wavelength Focusing in Inhomogeneous Media with a Metasurface Near Field Plate"

_sensors, 2019, doi:10.3390/s19204534_

Round 1

Reviewer 1 Report

In the manuscript titled ‘Sub-Wavelength Focusing in Inhomogeneous Media with a Metasurface Near Field Plate’, the authors presented an approach to design a subwavelength-focusing structure based on near field plate. Their approach fully accounts the interactions between the elements of the near field plate and the object of interest, which are difficult to model analytically using conventional approaches. The results are clearly presented and the paper is well written. I recommend it for publication provides the following comments are replied.

(1) The design problem considered in the current work is an inverse design problem. There has been intensive development in using neural network or adjoint method for inverse design recently, which are not mentioned and discussed in current manuscript.

(2)  The structures considered in the current work can also be modelled as coupled resonances. Although it is very difficult to analytically model the interaction between the resonances in inhomogeneous environment, it’s still possible to directly calculate the interaction from the radiation field of the resonances with the presence of inhomogeneous environment, which can be obtained from numerical simulations such FEM simulation. With that being said, how does the proposed approach compare with coupled-resonance based model?

Author Response

In the manuscript titled ‘Sub-Wavelength Focusing in Inhomogeneous Media with a Metasurface Near Field Plate’, the authors presented an approach to design a subwavelength-focusing structure based on near field plate. Their approach fully accounts the interactions between the elements of the near field plate and the object of interest, which are difficult to model analytically using conventional approaches. The results are clearly presented and the paper is well written. I recommend it for publication provides the following comments are replied.

(1) The design problem considered in the current work is an inverse design problem. There has been intensive development in using neural network or adjoint method for inverse design recently, which are not mentioned and discussed in current manuscript.

We have added references to machine learning and adjoint method approaches in the introduction (lines 40-42).

(2)  The structures considered in the current work can also be modelled as coupled resonances. Although it is very difficult to analytically model the interaction between the resonances in inhomogeneous environment, it’s still possible to directly calculate the interaction from the radiation field of the resonances with the presence of inhomogeneous environment, which can be obtained from numerical simulations such FEM simulation. With that being said, how does the proposed approach compare with coupled-resonance based model?

The structure could potentially be modeled in such a fashion, as the reviewer suggests, but this would add in an additional layer of abstraction (e.g. compute the FEM model, then insert results into coupled resonance model). Because the presented approach achieves an optimized design in a single FEM we have not directly compared it to a coupled-resonance based model, although that could be an interesting topic for future work.

Reviewer 2 Report

The Article ‘Sub-Wavelength Focusing in Inhomogeneous Media with a Metasurface Near Field Plate’ by A. Strikwerda et al. reports on the modeling of corrugated metasurfaces acting as near field plates which can focus the electric field at the sub-wavelength scale in inhomogeneous background media. The model used for finite element methods calculation is described in detail providing examples of different geometries. This research fits with the scope of the journal and can be of interest for the community of people working on theory and simulation of near field maps induced by metasurfaces.  However, the possible applications of the method developed here should be better pointed out in the introduction and in the discussion/conclusions also by mentioning experimental cases that could benefit from the method. In parallel, I recommend to mention and cite in the introduction relevant experimental works dealing with near field imaging/sensing at the subwavelength scale in condensed matter or biology.

Overall, I think the paper is good but has to be revised in order to be suitable for publication in Nanosensor.

In particular, I have a few more suggestions, which the authors should consider before resubmitting the paper.

1)line 18: The near-field research at the sub-wavelength scale is here poorly referenced. I suggest to add references such as

- Lezec, Henri J., et al. "Beaming light from a subwavelength aperture." Science 297.5582 (2002): 820-822.

- Luo, Chiyan, et al. "Subwavelength imaging in photonic crystals." Physical Review B 68.4 (2003): 045115.

- Rogers, Edward TF, et al. "A super-oscillatory lens optical microscope for subwavelength imaging." Nature materials 11.5 (2012): 432.

2) Lines 19-20: ‘Interest in such possibilities has renewed in recent decades with the advent of the perfect lens, the superlens, and near field plates’

I think the subwavelength near-field imaging techniques based on nanoscale tips or novel aperture probe devices should be also mentioned here. I suggest here some references:

- F. Keilmann, et al. "Near-field nanoscopy by elastic light scattering from a tip." Nano-optics and near-field optical microscopy (2009): 235-265.

- M.C. Giordano et al. "Phase-resolved terahertz self-detection near-field microscopy." Optics express 26.14 (2018): 18423-18435.

- R. Degl’Innocenti, et al. "Terahertz nanoscopy of plasmonic resonances with a quantum cascade laser." ACS Photonics 4.9 (2017): 2150-2157.

- M.C. Giordano, et al. "Phase-sensitive terahertz imaging using room-temperature near-field nanodetectors." Optica 5.5 (2018): 651-657.

3)line 23: If you consider also the scattering probe techniques spatial resolution of the order of  λ/1000  can be achieved. The λ/15 limit is more related to the near field techniques based on aperture probes. This difference should be pointed out in the text.  

4)Figure 1b: Typos and label shift in the x-scale

5)line 84-85: a TEM wave’ the acronym should be defined

6)lines 93-94:’ the support of Hz,foc (ky) in the the transverse wavenumber domain must exceed the free-space wavenumber k0.’

The sentence is not clear. Please specify better what you mean as 'support of Hz,foc in the transverse wavenumber domain' and why this should exceed the free space wavenumber.

There is a typo to correct: ‘the’ repeated twice

7)Lines 99-100: the sentence is not clear. Please clarify which function you consider for evaluating λ /10 FWHM and if this parameter corresponds to the spatial resolution you would like to have.

8)Line 136: ‘which dramatically increases the sensitivity’. I would expect the opposite effect (i.e. lower sensitivity for large grooves). Can you please clarify?

9)Figures 5b,c: The fields maps in b,c shows a degree of asymmetry of the magnetic field distribution along the y distribution (blue spots in the upper part of the maps not present in the bottom part). Since the system is y-symmetric I would expect a symmetric near-field distribution as well. Can you comment on that?

The intensity of the magnetic field in panel b,c should be provided. Is it the color scale the same in the two cases or not?

10) Caption of Fig.5 b,c. I suggest you to change as : ‘(b,c) Spatial distribution of the magnetic field, H, for Δy=0.65 and Δy=0, respectively.’

The comment: ‘Note that the groove depths have changed, but the focused field remains the same.’  should be integrated in the main text.

11)Line 323: ‘Some deviations can be seen at ρ/λ= 0.04’

The radius was previously indicated as r at line 302. If you consider the same quantity please unify the notation, otherwise you should define a new variable explicitly.

12)Line 326-327:’ and estimates the Ohmic heating that would occur within the targeted volume.’

It is not clear how the Ohmic heating has been estimated in panel c). Can you please comment on this point?

Can you also declare which quantity has been plotted in panel c)? Can you provide a color scale?

The caption of Fig.7c should be also clarified accordingly.

13) Figure 8c: the color scale of the logarithmic plot should be provided

14)Line 395: The position of the focal plane and of the tumor should be highlighted in fig.10a) and the x-coordinate specified in the main text .

15)Lines 395-396: ‘As can be seen the temperature in the focal plane is warmest near the tumor, where the electric field is greatest’. The English of this sentence should be revised.

16)Lines 401-402: The possible applications of your method should be better pointed out considering the recent experimental research in the field. Can you please provide few references of experimental activities related to your research or cases which would benefit by your method?

17)’ This method was shown to generate the desired focus in inhomogeneous environments, whereas the design optimized for free-space operation had its focus completely eliminated.’

The sentence is not clear, can you please clarify?

Author Response

The Article ‘Sub-Wavelength Focusing in Inhomogeneous Media with a Metasurface Near Field Plate’ by A. Strikwerda et al. reports on the modeling of corrugated metasurfaces acting as near field plates which can focus the electric field at the sub-wavelength scale in inhomogeneous background media. The model used for finite element methods calculation is described in detail providing examples of different geometries. This research fits with the scope of the journal and can be of interest for the community of people working on theory and simulation of near field maps induced by metasurfaces.  However, the possible applications of the method developed here should be better pointed out in the introduction and in the discussion/conclusions also by mentioning experimental cases that could benefit from the method. In parallel, I recommend to mention and cite in the introduction relevant experimental works dealing with near field imaging/sensing at the subwavelength scale in condensed matter or biology.

Overall, I think the paper is good but has to be revised in order to be suitable for publication in Nanosensor.

In particular, I have a few more suggestions, which the authors should consider before resubmitting the paper.

1) Line 18: The near-field research at the sub-wavelength scale is here poorly referenced. I suggest to add references such as

- Lezec, Henri J., et al. "Beaming light from a subwavelength aperture." Science 297.5582 (2002): 820-822.

- Luo, Chiyan, et al. "Subwavelength imaging in photonic crystals." Physical Review B 68.4 (2003): 045115.

- Rogers, Edward TF, et al. "A super-oscillatory lens optical microscope for subwavelength imaging." Nature materials 11.5 (2012): 432.

We have added several of the suggested references in line 19.

2) Lines 19-20: ‘Interest in such possibilities has renewed in recent decades with the advent of the perfect lens, the superlens, and near field plates’

I think the subwavelength near-field imaging techniques based on nanoscale tips or novel aperture probe devices should be also mentioned here. I suggest here some references:

- F. Keilmann, et al. "Near-field nanoscopy by elastic light scattering from a tip." Nano-optics and near-field optical microscopy (2009): 235-265.

- M.C. Giordano et al. "Phase-resolved terahertz self-detection near-field microscopy." Optics express 26.14 (2018): 18423-18435.

- R. Degl’Innocenti, et al. "Terahertz nanoscopy of plasmonic resonances with a quantum cascade laser." ACS Photonics 4.9 (2017): 2150-2157.

- M.C. Giordano, et al. "Phase-sensitive terahertz imaging using room-temperature near-field nanodetectors." Optica 5.5 (2018): 651-657.

The reviewer is correct, and we have added a brief discussion on lines 24-26.

3) Line 23: If you consider also the scattering probe techniques spatial resolution of the order of  λ/1000  can be achieved. The λ/15 limit is more related to the near field techniques based on aperture probes. This difference should be pointed out in the text. 

As mentioned above, the reviewer is correct to point out this issue and it is important to emphasize the differences between our approach vs using probes or apertures. We have added a brief discussion on lines 24-26.

4) Figure 1b: Typos and label shift in the x-scale

We do not see any typos in this caption, can the editor or reviewer comment on this? We have updated the scale in Fig 1b from meters to cm.

5) Line 84-85: ‘a TEM wave’ the acronym should be defined

Added to line 91.

6) Lines 93-94:’ the support of Hz,foc (ky) in the the transverse wavenumber domain must exceed the free-space wavenumber k0.’

The sentence is not clear. Please specify better what you mean as 'support of Hz,foc in the transverse wavenumber domain' and why this should exceed the free space wavenumber.

There is a typo to correct: ‘the’ repeated twice

The double "the" has been corrected. The sentence has been updated to make it clear that evanescent waves must be included, and therefore the transverse wave numbers must exceed that of the free space wave number (lines 100-102).

7) Lines 99-100: the sentence is not clear. Please clarify which function you consider for evaluating λ /10 FWHM and if this parameter corresponds to the spatial resolution you would like to have.

The function, a sinc, was listed in the previous sentence. In that sentence we have added explicit references to Hz for both the real and Fourier domain to clarify this. We have also updated the linking text to make it more clear that the FWHM is related to the previously mentioned sinc function (lines 106-109).

8) Line 136: ‘which dramatically increases the sensitivity’. I would expect the opposite effect (i.e. lower sensitivity for large grooves). Can you please clarify?

This is because of the asymptotic behavior of the tangent function in eq 4. We have added an explicit reference to that function in the sentence (lines 145-146).

9) Figures 5b,c: The fields maps in b,c shows a degree of asymmetry of the magnetic field distribution along the y distribution (blue spots in the upper part of the maps not present in the bottom part). Since the system is y-symmetric I would expect a symmetric near-field distribution as well. Can you comment on that?

The intensity of the magnetic field in panel b,c should be provided. Is it the color scale the same in the two cases or not?

Thank you for pointing this out. The issue has been corrected.

10) Caption of Fig.5 b,c. I suggest you to change as : ‘(b,c) Spatial distribution of the magnetic field, H, for Δy=0.65 and Δy=0, respectively.’

This has been changed as suggested.

The comment: ‘Note that the groove depths have changed, but the focused field remains the same.’ should be integrated in the main text.

Updated in lines 282-283.

11) Line 323: ‘Some deviations can be seen at ρ/λ= 0.04’

The radius was previously indicated as r at line 302. If you consider the same quantity please unify the notation, otherwise you should define a new variable explicitly.

We have updated to rho in line 313 and 382, and added an explicit mention of the cylindrical coordinate system used in lines 300 - 301.

12) Line 326-327:’ and estimates the Ohmic heating that would occur within the targeted volume.’

It is not clear how the Ohmic heating has been estimated in panel c). Can you please comment on this point?

This is from J dot E, which we have included in the text in line 340.

Can you also declare which quantity has been plotted in panel c)? Can you provide a color scale?

Color bars have been added and the caption has been updated accordingly.

The caption of Fig.7c should be also clarified accordingly.

Done, as mentioned above.

13) Figure 8c: the color scale of the logarithmic plot should be provided

This has been added.

14) Line 395: The position of the focal plane and of the tumor should be highlighted in fig.10a) and the x-coordinate specified in the main text.

The caption has been updated to indicate that the focal plane and tumor are shown with black lines in the figure.

15) Lines 395-396: ‘As can be seen the temperature in the focal plane is warmest near the tumor, where the electric field is greatest’. The English of this sentence should be revised.

Updated in lines 409-410.

16) Lines 401-402: The possible applications of your method should be better pointed out considering the recent experimental research in the field. Can you please provide few references of experimental activities related to your research or cases which would benefit by your method?

This is a good point. While we are not aware of directly comparable work, we have listed two similar applications that could benefit from tailoring of the near-field to account for inhomogenous environments (wireless power transfer and crack detection). This update is on lines 415-417.

17) This method was shown to generate the desired focus in inhomogeneous environments, whereas the design optimized for free-space operation had its focus completely eliminated.’

The sentence is not clear, can you please clarify?

This was on line 414-415 in the original text, and was referring to section 4.1 NFP With Grooves. In that section we showed how a NFP designed for free space would only work appropriately in free space, but failed miserably at focusing inside of an inhomogenous environment. To make this more clear, we have updated the sentence the reviewer mentions (lines 427-429) and also revised section 4.1 to make this more apparent (lines 309-313 in the original text vs lines 320-326 in the updated text).

Reviewer 3 Report

The authors present a simulated development of a corrugated near field plate (NFP) in order to generate a sub-wavelength focus in inhomogeneous media.

That is important since it deals with two major contributions:

1) Most important from this reviewer's point of view: the extension of previous studies from Grbic et at to an inhomogeneous media.

2) Secondly: working in sub-wavelength focus allows overcoming the diffraction limit.

In addition the paper is written in a well organized and clear English.

For those reasons I consider the paper should be published nearly in its present state except some minor remarks.

a) Different acronyms are used along the text and their meaning are explained except BOLD (line 45). Sorry but I did not undertand

b) The dimensions of the grooves should be clearly introduced in a picture or in a sketch. This will help the understanding of the paper: i.e w (width), a (separation) and length of the grooves.

Finally, and most important, from the explanation in the introduction, it seems that the most important contribution is the extension of the work from Grbic et al to inhomogeneous media. Is that so? In case so I miss a further explanation of what are the difficulties rising (in the analysis and in the simulation fields) from making this extension. Later on the improvements on making this effort is clearly explained but the difficulties found should be clearly emphasized.

Author Response

The authors present a simulated development of a corrugated near field plate (NFP) in order to generate a sub-wavelength focus in inhomogeneous media.

That is important since it deals with two major contributions:

1) Most important from this reviewer's point of view: the extension of previous studies from Grbic et at to an inhomogeneous media.

2) Secondly: working in sub-wavelength focus allows overcoming the diffraction limit.

In addition the paper is written in a well organized and clear English.

For those reasons I consider the paper should be published nearly in its present state except some minor remarks.

a) Different acronyms are used along the text and their meaning are explained except BOLD (line 45). Sorry but I did not undertand

BOLD stands for blood-oxygen level dependent and we have included that in line 50.

b) The dimensions of the grooves should be clearly introduced in a picture or in a sketch. This will help the understanding of the paper: i.e w (width), a (separation) and length of the grooves.

The dimensions have been added to the figure caption.

Finally, and most important, from the explanation in the introduction, it seems that the most important contribution is the extension of the work from Grbic et al to inhomogeneous media. Is that so? In case so I miss a further explanation of what are the difficulties rising (in the analysis and in the simulation fields) from making this extension. Later on the improvements on making this effort is clearly explained but the difficulties found should be clearly emphasized.

This is correct. The most important contribution is the application of near field plates to inhomogenous environments, and it is this extension that makes the remainder of the work possible. We have changed line 65-66 to further emphasize this point in the introduction, as well as emphasized later (line 70) that previous work operate "solely" in free-space settings.